# Climate Indices-Based Analysis of Rainfall Spatiotemporal Variability in Pernambuco State, Brazil

**Tarciana Rafaela Barbosa Figueiroa Silva [1], Carlos Antonio Costa dos Santos [1,\*], Delson José Figueiroa Silva [1], Celso Augusto Guimarães Santos [2], Richarde Marques da Silva [3] and José Ivaldo Barbosa de Brito [1]**

[1] Graduate Program in Meteorology, Academic Unity of Atmospheric Sciences, Federal University of Campina Grande, Campina Grande 58109-970, Paraíba, Brazil; tarc1an4@gmail.com (T.R.B.F.S.); arutanissa@hotmail.com (D.J.F.S.); jose.ivaldo@ufcg.edu.br (J.I.B.d.B.)

[2] Department of Civil and Environmental Engineering, Federal University of Paraíba, João Pessoa 58051-900, Paraíba, Brazil; celso@ct.ufpb.br

[3] Department of Geosciences, Federal University of Paraíba, João Pessoa 58051-900, Paraíba, Brazil; richarde@geociencias.ufpb.br

\* Correspondence: carlos.santos@ufcg.edu.br

**Abstract:** In Brazil, the Northeast region, particularly the state of Pernambuco, is prone to extreme hydrological events, especially episodes of heavy rain and long periods of drought. This study examined the spatiotemporal variability of rainfall in Pernambuco and the occurrence of extreme rainfall events. In this study, the following climate indices were used in daily rainfall data from the historical series from 1990 to 2020: (a) the Precipitation Concentration Index (*PCI*), (b) the Standardized Precipitation Index (*SPI*), (c) the Precipitation Concentration Degree (*PCD*), (d) the Precipitation Concentration Period (*PCP*), and (e) the Rainfall Anomaly Index (*RAI*), and the non-parametric Mann–Kendall and Sen's slope tests, for the analysis of trends in the series of precipitation and the studied climate indices. The results obtained indicated that the east of the state presents more distributed precipitation throughout the year, being "moderately seasonal" in the mesoregions Metropolitana do Recife and Zona da Mata; "seasonal" in the Agreste Pernambucano mesoregion; and "strongly seasonal" in the Sertão Pernambucano and Sertão do São Francisco mesoregions. The *SPI* found that the most severe and extreme droughts occurred in almost all mesoregions during the 1990s, except in the Sertão do São Francisco mesoregion, where they were predominant in the 2010s. Furthermore, based on the *RAI* index, it was possible to have a greater occurrence of negative anomalies than positive anomalies, especially in the last decade (2010–2020), indicating a possible change in rainfall patterns. However, more in-depth research is required to determine the possible causes of this increased number of negative anomalies. Finally, the trend analysis indicated that only the *SPI-12* presented trends in the Metropolitana do Recife, Agreste, and Sertão Pernambucano mesoregions. However, Sen's slope test indicated that the magnitude of these trends is not significant.

**Keywords:** climate change; anomalies; extreme climatic events; northeast region; Brazil

## 1. Introduction

The planet's climate is constantly transformed by its dynamics or by the intensification of human activities, resulting in changes in climate patterns that directly affect natural ecosystems, extreme events, and socio-economic systems. In general, there is a consensus among most researchers that the gradual increase in GHG concentrations, mainly $CO_2$, intensifies the greenhouse effect, resulting in an irreversible warming process, affecting, in an unprecedented way, natural systems and humans [1–6]. According to the IPCC (2021) [7], the human influence on global warming, mainly through the burning of fossil fuels and deforestation, is unequivocal and unquestionable, with widespread and rapid changes in the atmosphere, ocean, cryosphere, and biosphere [8]. Thus, several

studies indicate that the increase in the average surface temperature may lead to an acceleration of the hydrological cycle, resulting in significant changes in precipitation patterns in different parts of the globe, favoring the occurrence of extreme precipitation and, consequently, more intense droughts and frequent [9–13].

The effects of climate change are already visible in several regions of the planet, especially in underdeveloped and developing countries, which, although not responsible for the highest greenhouse gas emissions, are more vulnerable to climate change [14]. They are usually found in areas of greater exposure to extreme events, with high rainfall variability, and are most often heavily dependent on agriculture, the most climate-sensitive sector of the economy. Furthermore, they usually have socio-economic weaknesses that reduce their ability to deal with such impacts [15,16].

Extreme events have become increasingly frequent worldwide, worrying authorities and scientific communities due to their large-scale impacts. An event is said to be extreme when it presents anomalies concerning climatology. From the specific point of view of rainfall, these events have their rainfall totals in a certain period deviation of great magnitude from their usual conditions. They are characterized by their high intensity, low frequency (rare and uncommon), and great potential to trigger disasters, causing significant socio-economic and environmental damage [17,18].

In Brazil, extreme hydrological events represent a significant threat to the Northeast region since it presents high spatial and temporal variability in its rainfall regime [19]. The state of Pernambuco, which is located in this region, is susceptible to extreme rainfall events and is frequently affected by heavy rains, which cause coastal erosion, landslides, floods, and inundations, especially in the Metropolitana do Recife, Zona da Mata, and Agreste Pernambucano mesoregions [20,21], and by cycles of recurrent droughts, which mainly affect the semiarid region [22]. In the last 20 years, floods, inundations, and landslides were the disasters that most displaced people and caused the death of the population of Pernambuco. An example of this vulnerability was observed in 2010 and 2011, when heavy rainfall severely affected several municipalities in the state, causing massive flooding and destroying entire cities [23,24]; and in 2022, when heavy rains, combined with the lack of urban planning, caused landslides in the Metropolitan Region of Recife, causing the death of hundreds of people and leaving thousands homeless, being considered until then the biggest disaster of the 21st century in the state [25,26]. Concerning droughts, according to data from CEPED (2013) [24], between 1991 and 2012, there were 1308 official records of drought and drought in Pernambuco, where of the 185 municipalities in the state, 172 were affected by this type of disaster, resulting in socio-economic and environmental damage of great magnitude, and also according to Brito et al. (2018) [27] and Vieira et al. (2021) [28], the drought event that occurred from 2012–2017 was considered the worst in the last 30 years. It is worth mentioning that this phenomenon increases from east to west as municipalities move away from the Atlantic Ocean, with the coast and Zona da Mata being the least affected regions (CEPED, 2013) [24].

Thus, due to the impacts caused by the irregularity of the rains, it is necessary to carry out studies to investigate the magnitude and trends of these critical events. Considering that in Brazil, especially the state of Pernambuco, there are no such comprehensive studies on precipitation, the objective of this study was to analyze the spatial and temporal variability of rainfall in the state of Pernambuco and the occurrence of extreme events in the period from 1990 to 2020, using the following climate indices: the Precipitation Concentration Index (*PCI*) [29], the Standardized Precipitation Index (*SPI*) [30], the Precipitation Concentration Degree (*PCD*), and the Period of Precipitation Concentration (*PCP*) [31], and the Rainfall Anomaly Index (*RAI*) [32]. These indices help to understand the climatology and rainfall in the region, helping strategic planning and decision making in the most diverse sectors of society, such as meteorology, water resources, supply, agriculture, livestock, health, and urban planning.



## 2. Material and Methods

### 2.1. Study Area and Dataset

The study area is the state of Pernambuco, located in the east-central region of northeastern Brazil, more precisely in coordinates −7°15′ and −9°27′ latitude and −34° and −48°19′ longitude. It occupies an area of 98,067,880 km². The state is geographically divided into five mesoregions: Metropolitana do Recife, Zona da Mata, Agreste Pernambucano, Sertão Pernambucano, and Sertão do São Francisco (Figure 1). According to SRHE (2013) [33], with approximately 70% of its territory being semiarid climates, the state of Pernambuco has distinct climatic characteristics that vary significantly according to the distance from the sea and altitude, reaching the dry region in the Sertão do São Francisco mesoregion. Along the coast and the Zona da Mata region, the climate varies between humid and sub-humid, while the regions in the Agreste and Northwest of the state have a climate ranging between sub-humid and dry sub-humid. Concerning rainfall, the state of Pernambuco is characterized by a marked interannual variability, both in spatial and temporal scales, thus constituting one of the most striking characteristics of the state's climate [34,35]. This is due to the influence of SST variability in the Equatorial Pacific and Tropical Atlantic with anomalous phases in each of the oceans, corresponding to El Niño, La Niña, and SST configurations Anomaly Gradient in the Tropical Atlantic [36]. These configurations determine the performance and intensity of the various meteorological systems that operate in the state that, depending on their phase, act either intensifying or weakening the rainfall patterns [37,38]. According to SECTMA (1998) [39], the main meteorological systems responsible for rainfall in the state of Pernambuco are the Intertropical Convergence Zone (ITCZ) [40,41], the Eastern Wave Disturbances [42,43], the High-Level Cyclonic Vortices [44–46], the Cold Fronts [40,44,47–49], in addition to sea and land breeze systems.

Daily rainfall data from 1990 to 2020 were obtained for 38 weather stations across the state of Pernambuco (Table 1) from the Pernambuco Water and Climate Agency (APAC) website (http://old.apac.pe.gov.br/meteorologia/monitoramento-pluvio.php (accessed on 15 January 2021)) and the National Institute of Meteorology (Inmet) (https://bdmep.inmet.gov.br/ (accessed on 25 February 2021)). The meteorological stations used in this work underwent a quality control where the consistency of daily precipitation values was evaluated. All precipitation data were properly treated, and stations with incorrect data and/or outliers were removed, with the highest data record, most common periods, and few failures being prioritized (Table 1).

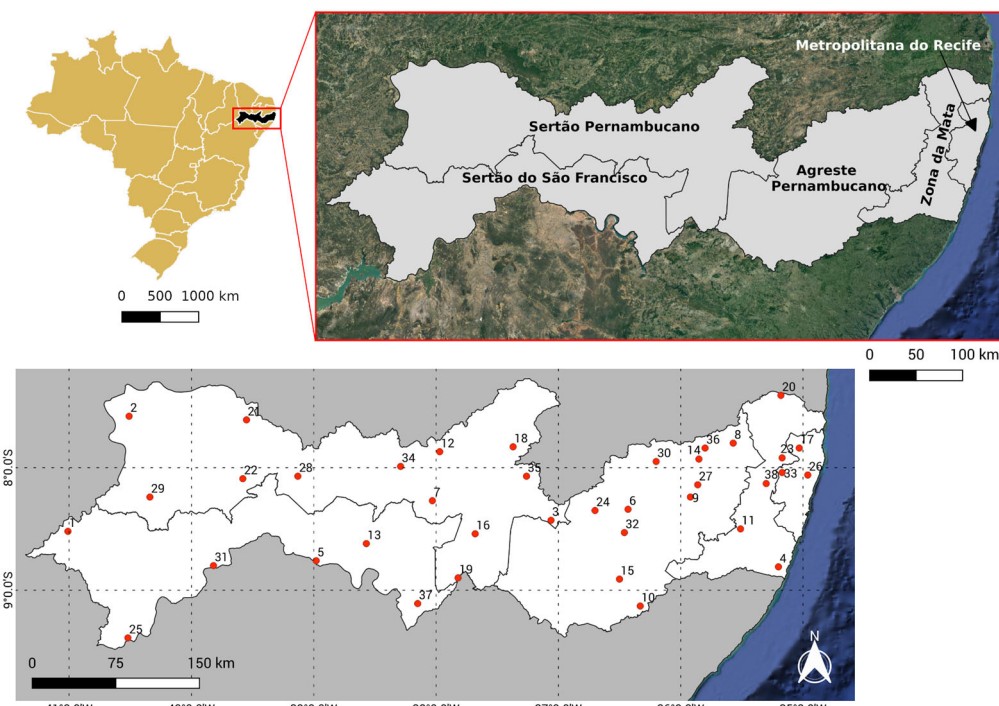

**Figure 1.** Map of Brazil highlighting Pernambuco and the locations of the meteorological stations used in this study.

**Table 1.** Selected rain stations and their respective geographic coordinates.

| No. | Rain Gauge Station | Mesoregion | Operator | Latitude | Longitude | Altitude |
|---|---|---|---|---|---|---|
| 1 | Afrânio | SFM | APAC | −8.52 | −41.01 | 522.9 |
| 2 | Araripina | SPM | APAC | −7.58 | −40.51 | 639.5 |
| 3 | Arcoverde | SPM | INMET | −8.43 | −37.06 | 683.91 |
| 4 | Barreiros | ZMM | APAC | −8.81 | −35.2 | 13.9 |
| 5 | Belém do São Francisco | SFM | APAC | −8.76 | −38.98 | 309 |
| 6 | Belo Jardim | APM | APAC | −8.34 | −36.43 | 612.8 |
| 7 | Betânia | SPM | APAC | −8.27 | −38.03 | 440.4 |
| 8 | Bom Jardim | APM | APAC | −7.8 | −35.57 | 342.8 |
| 9 | Caruaru (IPA) | APM | APAC | −8.24 | −35.92 | 608.5 |
| 10 | Correntes | APM | APAC | −9.13 | −36.33 | 399.1 |
| 11 | Cortês | ZMM | APAC | −8.5 | −35.51 | 195.2 |
| 12 | Flores | SPM | APAC | −7.87 | −37.97 | 480.9 |
| 13 | Floresta | SFM | APAC | −8.62 | −38.57 | 318.1 |
| 14 | Frei Miguelinho | APM | APAC | −7.93 | −35.85 | 396 |
| 15 | Garanhuns | APM | INMET | −8.91 | −36.5 | 827.78 |
| 16 | Ibimirim | SPM | APAC | −8.54 | −37.68 | 395 |
| 17 | Igarassu | MRM | APAC | −7.84 | −35.03 | 56 |
| 18 | Iguaraci | SPM | APAC | −7.83 | −37.37 | 590.1 |
| 19 | Inajá | SPM | APAC | −8.9 | −37.82 | 359 |
| 20 | Itambé | ZMM | APAC | −7.41 | −35.18 | 142.8 |
| 21 | Moreilândia | SPM | APAC | −7.61 | −39.55 | 502 |
| 22 | Parnamirim | SPM | APAC | −8.09 | −39.58 | 392.1 |
| 23 | Paudalho | ZMM | APAC | −7.92 | −35.17 | 167.1 |
| 24 | Pesqueira | APM | APAC | −8.35 | −36.7 | 667.9 |
| 25 | Petrolina | SFM | INMET | −9.39 | −40.52 | 372.54 |
| 26 | Recife | MRM | INMET | −8.06 | −34.96 | 11.3 |

| 27 | Riacho das Almas | APM | APAC | −8.14 | −35.86 | 410.2 |
| 28 | Salgueiro | SPM | APAC | −8.07 | −39.13 | 444.3 |
| 29 | Santa Cruz da Verenada | SPM | APAC | −8.24 | −40.34 | 514.1 |
| 30 | Santa Cruz do Capibaribe | APM | APAC | −7.95 | −36.2 | 446.5 |
| 31 | Santa Maria da Boa Vista | SFM | APAC | −8.8 | −39.82 | 361 |
| 32 | São Bento do Una | APM | APAC | −8.53 | −36.46 | 617.3 |
| 33 | São Lourenço da Mata | MRM | APAC | −8.04 | −35.17 | 87 |
| 34 | Serra Talhada | SPM | APAC | −7.99 | −38.29 | 437.5 |
| 35 | Sertânia | SPM | APAC | −8.07 | −37.26 | 555 |
| 36 | Surubim | APM | INMET | −7.84 | −35.8 | 421.44 |
| 37 | Tacaratu | SFM | APAC | −9.11 | −38.15 | 518.2 |
| 38 | Vitória de Santo Antão | ZMM | APAC | −8.13 | −35.3 | 166.7 |

Notes: APM: Agreste Pernambucano; MRM: Metropolitana do Recife; SFM: Sertão do São Francisco; SPM: Sertão Pernambucano; ZMM: Zona da Mata.

### 2.2. Rainfall Indices

#### 2.2.1. The PCI

The *PCI* was initially proposed by Oliver [29] and later developed by De Luis et al. (1997) [50] and is an index that allows for determining the degree of concentration or seasonality of rainfall [51,52] based on its monthly variability, making it possible to detect changes in the weather patterns of a given location, as well as investigating the risks of extreme events for this variable [53,54]. The following equation defines the *PCI*:

$$PCI = 100 \cdot \frac{\sum_{i=1}^{12} p_i^2}{\left(\sum_{i=1}^{12} p_i\right)^2} \tag{1}$$

where $p_i$ is the monthly rainfall in the month $i$.

According to Oliver (1980) [29], uniform rainfall distribution is indicated by values less than or equal to 10, and as this value increases, the more concentrated the rainfall becomes. Therefore, the seasonal concentration of rainfall is classified according to Table 2.

**Table 2.** Precipitation Concentration Index classification.

| PCI | Seasonal Concentration |
| --- | --- |
| $PCI \leq 10$ | Uniform |
| $10 < PCI < 15$ | Moderately seasonal |
| $16 < PCI \leq 20$ | Seasonal |
| $PCI > 20$ | Strongly seasonal |

#### 2.2.2. The SPI

The *SPI* is an index whose main characteristic is the possibility of quantifying the deficit or excess of rainfall in different temporal scales and being one of the most used in the world. Blain (2014) [55] defines the *SPI* as a mathematical algorithm to detect and characterize rainfall anomalies related to an expected weather condition. Thus, negative *SPI* values indicate below-normal rainfall, while positive values reflect above-average rainfall. Since it was proposed, the *SPI* has been widely used in several studies in different parts of the world, such as Hungary [56], Portugal [57], Lebanon [58], Turkey [59], Greece [60], Brazil [27,61,62], Ethiopia [63], Romania [64], and Togo [65].

The *SPI* has been commonly used to monitor drought conditions or excess precipitation. Its use is strongly recommended by the World Meteorological Organization (WMO) since it makes it possible to compare different regions and climates. In addition, to calculate this index, the first step is to find the probability density function that best fits rainfall

frequency distribution on the desired time scale. Among the various distributions suggested by the literature, in this study, the gamma distribution was used to represent the theoretical distribution of this variable since it has been widely used in climatology. In order to verify the adherence of the rainfall time series to the gamma probability distribution, the Kolmogorov–Smirnov test was used at a significance level of 5%, where the following hypotheses H0 were considered: the gamma distribution fits the data series and H1: the gamma distribution does not fit the data series.

The gamma distribution is defined by the probability density function as:

$$g(x) = \frac{1}{\Gamma(\alpha)\beta\alpha} \cdot x^{\alpha-1} e^{\frac{-x}{\beta}} \tag{2}$$

where $\alpha > 0$ is the shape parameter, $\beta > 0$ is the scale parameter, and $x > 0$ is the amount of rainfall. The gamma function is given by:

$$\Gamma(\alpha) = \int_0^\infty y^{\alpha-1} e^{-y} dy \tag{3}$$

To estimate the parameters $\alpha$ and $\beta$, Thom's maximum likelihood method [66] is used, according to Equations (4)–(6):

$$\alpha = \frac{1}{4A}\left(1 + \sqrt{1 + \frac{4A}{3}}\right) \tag{4}$$

$$A = ln(\bar{x}) - \frac{\sum_i^n x_i}{n} \tag{5}$$

$$\beta = \frac{\bar{x}}{\alpha} \tag{6}$$

where $n$ is the number of observations in the sample, $\bar{x}$ is the mean value of precipitation, and $x_i$ are the observed precipitations. With the resulting parameters, the cumulative probability $G(x)$ of a rainfall event was calculated for the desired time scale according to Equation (7):

$$G(x) = \int_0^x g(x)dx = \frac{1}{\beta^\alpha \Gamma(\alpha)} \int_0^x x^{\alpha-1} e^{\frac{-x}{\beta}} dx \tag{7}$$

However, it is worth noting that the gamma function is not defined for $x = 0$. As the historical series of rainfall can contain zeros, the cumulative probability looks like this:

$$H(x) = q + (1 - q)G(x) \tag{8}$$

where $q$ is the probability of null values in the analyzed sample, which is calculated as $q = m/n$, where $m$ represents the number of events whose rainfall was equal to zero and $n$ is the number of observations. Finally, the cumulative probability $H(x)$ is converted into a normalized random variable ($Z$), which obeys a normal distribution of mean 0 and standard deviation 1, resulting in the $SPI$ value, according to Equations (10) and (11). Table 3 presents the $SPI$ classification for dry and rainy periods.

$$Z = SPI = -\left(t - \frac{c_0 + c_1 t + c_2 t^2}{1 + d_1 t + d_2 t^2 + d_3 t^3}\right) \text{ and } t = \sqrt{ln\left(\frac{1}{(H(x))^2}\right)}, \text{ for } 0 < H(x) \leq 0.5 \tag{9}$$

$$Z = SPI = +\left(t - \frac{c_0 + c_1 t + c_2 t^2}{1 + d_1 t + d_2 t^2 + d_3 t^3}\right) \text{ and } t = \sqrt{ln\left(\frac{1}{(1-H(x))^2}\right)}, \text{ for } 0.5 < H(x) \leq 1.0 \tag{10}$$

where $c_0 = 2.515517$, $c_1 = 0.802853$, $c_2 = 0.010328$, $d_1 = 1.432788$, $d_2 = 0.189269$, and $d_3 = 0.001308$.

**Table 3.** Standardized Precipitation Index classification.

| SPI | Classification |
|---|---|
| ≥2.00 | Extreme wet |
| 1.99 to 1.50 | Severe wet |
| 1.49 to 1.00 | Moderate wet |
| 0.99 to 0.50 | Mild wet |
| 0.49 to −0.49 | Normal |
| −0.50 to −0.99 | Mild drought |
| −1.00 to −1.49 | Moderate drought |
| −1.50 to −1.99 | Severe drought |
| ≤−2.00 | Extreme drought |

In order to verify the most significant events in Pernambuco State from 1990 to 2020, an average *SPI* was calculated on the scales of 3, 6 and 9 months for each mesoregion. The calculation of this average *SPI* consisted of calculating the *SPI* of each rain station in the time scales mentioned above. After this first step, the average of this index was calculated in each of the five mesoregions of the state.

2.2.3. The PCD and the PCP

The *PCD* and the *PCP* were two indices capable of quantitatively characterizing the concentration and degree of dispersion of rainfall throughout the year in a particular location [67]. The *PCD* is an index that expresses the degree of distribution of total rainfall over the 12 months of the year. The *PCP* represents the period (month) in which total rainfall was concentrated within a year. Since they were proposed, these indices have been widely used, mainly in Asia; however, studies using such indices in Brazil are still scarce.

In order to calculate the *PCD* and the *PCP*, it is assumed that the monthly rainfall data (r) are vectors oriented in the radial direction in a trigonometric circle, where the angles indicate the months of the year. Thus, using the decompositions of these vectors in the plane, Equations (12) and (13) can be expressed:

$$R_x = \sum_{i=1}^{12} r_{xi} = \sum_{i=1}^{12} r_i \; sen\theta_i \tag{11}$$

$$R_y = \sum_{i=1}^{12} r_{yi} = \sum_{i=1}^{12} r_i \; cos\,\theta_i \tag{12}$$

where *i* represents the month number of the year, $\theta$ is the angle associated with each month of the year, which can vary from 0° to 360° and $R_x$ and $R_y$ represent the sum of the vector projections that indicate monthly rainfall on the *x* and *y* axes, respectively. From Equations (12) and (13), the *PCD* and *PCP* values are given as follows:

$$PCD = \frac{\sqrt{\left(R_x^2 + R_y^2\right)}}{\psi}, \text{ where } \psi = \sum_{i=1}^{12} r_i \tag{13}$$

$$PCP = \tan^{-1}\left(\frac{R_x}{R_y}\right) \tag{14}$$

The *PCD* value varies between 0 and 1, where values close to 0 indicate rains more distributed throughout the year, while values close to 1 indicate rains concentrated in a short period. The *PCP* is given in degrees and indicates the month in which total rainfall was concentrated within a year. Table 4 presents the relationship between the *PCP* values and the corresponding month.

**Table 4.** Months corresponding to each *PCP* value.

| Month | Jan | Feb | Mar | Apr | May | Jun | Jul | Aug | Sep | Oct | Nov | Dec |
|-------|-----|-----|-----|-----|------|------|------|------|------|------|------|------|
| *PCP* | 0° | 30° | 60° | 90° | 120° | 150° | 180° | 210° | 240° | 270° | 300° | 330° |

### 2.2.4. RAI

Another parameter used in this work was the *RAI* to identify extreme anomalies of positive and negative order in the time series of rainfall and, in this way, classify dry and wet periods at different intensity levels. The main characteristic of this index is its procedural simplicity since it only needs rainfall data to be calculated, being suitable for use in semiarid and/or tropical regions, mainly for the *NEB* region [68,69].

To determine the *RAI*, we used Equations (15) and (16):

$$RAI = 3\left[\frac{(P-\bar{P})}{(\bar{M}-\bar{P})}\right] \text{ for positive anomalies} \tag{15}$$

$$RAI = -3\left[\frac{(P-\bar{P})}{(\bar{X}-\bar{P})}\right] \text{ for negative anomalies} \tag{16}$$

where $P$ is the observed rainfall (mm) in the year in which the *RAI* will be calculated, $\bar{P}$ is the average annual rainfall of the historical series (mm), and $\bar{M}$ and $\bar{X}$ represent the average of the ten highest and lowest annual rainfall values of the historical series (mm), respectively. From the values found, rainfall can be classified according to Table 5.

**Table 5.** Rainfall Anomaly Index rating.

| *RAI* | Intensity Class |
|-------|-----------------|
| ≥4.00 | Extreme wet |
| 3.00 to 3.99 | High wet |
| 2.00 to 2.99 | Moderate wet |
| 0.5 to 1.99 | Low wet |
| −0.49 to 0.49 | Normal |
| −1.99 to −0.5 | Mild drought |
| −2.00 to −2.99 | Moderate drought |
| −3.00 to −3.99 | High drought |
| ≤−4.00 | Extreme drought |

### 2.3. Trends

The non-parametric Mann–Kendall tests [70,71] were applied at a significance level of 5% to detect the presence or absence of a trend in the time series of rainfall and the climate indices used in this work, and the Sen's slope test [72], to estimate the magnitude of the trends found.

### 2.3.1. The Mann–Kendall Test

The Mann–Kendall test is a non-parametric test commonly used to verify whether a given historical series presents a statistically significant trend, and its use is strongly recommended by the WMO, mainly to identify trends in hydrological series [73–76]. The Mann–Kendall test is based on the following equation:

$$S = \sum_{k=1}^{n-1} \sum_{j=k+1}^{n} sgn\left(x_j - x_k\right) \tag{17}$$

where $n$ is the size of the time series, $x_j$ and $x_k$ are the data values in the time series $j$ and $k$, respectively, and $sgn(x_j - x_k)$ is the signal function defined as:

$$sgn(x) = \begin{cases} +1; \text{ if } x > 0 \\ 0; \text{ if } x = 0 \\ +1; \text{ if } x < 0 \end{cases} \tag{18}$$

The variance in the trend is calculated by:

$$Var(S) = \frac{n(n-1)(2n+5) - \sum_{j=1}^{p} t_j (t_{j-1})(2t_j + 5)}{18} \tag{19}$$

where $n$ is the number of observations and, considering that the series may have groups with equal observations, $p$ is the number of groups with equal observations, and $t_j$ is the number of equal observations in group $j$.

The $S$ and $Var(S)$ values are used to calculate the Z-test statistic as follows:

$$Z_{MK} = \begin{cases} \dfrac{S-1}{\sqrt{Var(S)}}; \; for \, S > 0 \\ 0 \; ; \; for \; S = 0 \\ \dfrac{S+1}{\sqrt{Var(S)}}; \; for \; S < 0 \end{cases} \tag{20}$$

The significance of the trend can be verified by comparing the absolute value of $Z_{MK}$ with that of the standardized normal variable for a given significance level $\alpha$ chosen. In the case $|Z_{MK}| > Z_{1-\frac{\alpha}{2}}$, the null hypothesis is rejected, and it is considered that there is a trend in the analyzed series.

2.3.2. Sen's Slope Estimator

The non-parametric test developed by Sen (1968) [72] estimates the slope of a trend (variation per unit of time), assuming that it is linear. This means that:

$$f(t) = Qt + B \tag{21}$$

where $f(t)$ is the function representing the evolution of the time series, which can increase or decrease, $t$ is in the unit of time, $Q$ is the slope, and $B$ is a constant.

In order to obtain the estimate of slope $Q$, the slopes of all pairs of data are calculated:

$$Q_i = \frac{x_j - x_k}{j - k}, for \, i = 1, 2 \ldots n \tag{22}$$

where $x_j$ and $x_k$ are the data values at time $j$ and $k$ ($j > k$), respectively.

If there are $n$ values $x_j$ in the dataset, we get $N = \frac{n(n-1)}{2}$ slope estimates $Q_i$. The $N$ values of $Q_i$ are ranked from smallest to largest, and the median of Sen's slope estimator ($Q_{med}$) is calculated as:

$$Q_{med} = Q_{\frac{(N+1)}{2}}; if \, N \, is \, odd; \; \frac{1}{2}\left( Q_{\left(\frac{N}{2}\right)} + Q_{\frac{(N+2)}{2}} \right); if \, N \, is \, even \tag{23}$$

The sign of $Q_{med}$ reflects the trend of the data, while the value indicates the slope of the trend [77].

**3. Results**

Figure 2a depicts the spatial distribution of the average annual rainfall in Pernambuco State from 1990 to 2020. When analyzing the figure, it is noted that the highest rainfall rates are concentrated in the Metropolitan Mesoregion of Recife and Zona da Mata with rainfall greater than 1200 mm/year and above 900 mm/year, respectively. In the Agreste Pernambucano mesoregion, almost all areas are 500 to 700 mm/year, with lower accumulations in small areas to the north of the mesoregion and accumulations greater than 800 mm/year in some strips to the south. The Sertão do São Francisco and Sertão mesoregions are the ones with the lowest values, with rainfall between 300 and 700 mm/year in the Sertão do São Francisco mesoregion and accumulated in the range of 400 to 700 mm/year in the Sertão Pernambucano mesoregion.

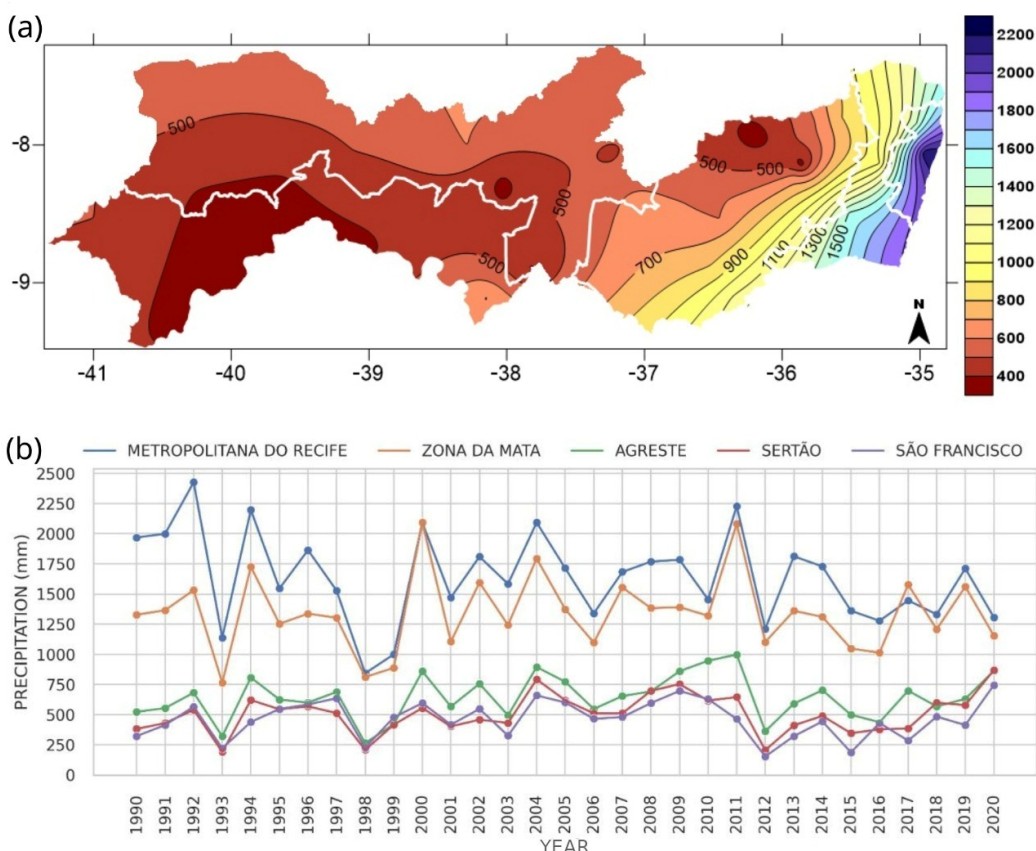

**Figure 2.** Spatial distribution (**a**) and average annual precipitation (**b**) in the Pernambuco mesoregions from 1990 to 2020.

The behavior of the average annual precipitation of each mesoregion of the state of Pernambuco is illustrated in Figure 2b. It is possible to observe the interannual variability throughout the time series. The highest averages of annual rainfall are observed in the Metropolitana of Recife and Zona da Mata mesoregions, where the years with the highest average rainfall were 1992, with an average volume of 2426.3 mm, and 2011, with an average of 2228.46 mm, in the Metropolitan area of Recife; the year 2000, with an average volume of 2091.98 mm, and 2011, with an average volume of 2082.8 mm, in the Zona da Mata. Regarding the years with the lowest average rainfall, it is observed that these occurred in the years 1998, with an average of 842.93 mm, and 1999, with an average volume of 999.5 mm, in the Metropolitan Mesoregion of Recife; and the years 1993, with an average of 763 mm, and 1998, with an average of 812.18 mm, in the Zona da Mata mesoregion. In the Agreste of the state, it can be seen that above-average rainfall occurred in the years 2011, 2010 and 2004, while the years 1998, 1993 and 2012 were characterized by being the three driest years of the entire historical series, with rainfall of less than 370 mm. The mesoregions that presented the lowest rainfall were Serto Pernambucano and Sertão do São Francisco, where the years with the lowest average rainfall were, respectively, 1993, when the average volume was 191.2 mm, and 2012, with an average of 206 mm, respectively, in the Sertão Pernambucano mesoregion; and 2012, with average values of 153.45 mm, and 2015, with an average volume of 188.2 mm, in the Sertão do São Francisco mesoregion. In addition, in both mesoregions, it is possible to observe that the year 2020 presented above-average precipitation, with an average volume of 865.75 mm in the Sertão Pernambucano and 745.3 mm in the Sertão do São Francisco mesoregion.

Analyzing Figure 3 and considering the average monthly variation of rainfall in two periods of six months each, it is observed that the rainy season in the mesoregions Metro-

politana do Recife (Figure 3a), Zona da Mata (Figure 3b), and Agreste Pernambucano (Figure 3c) occurred from March to August, where the months of June and July had the highest rainfall records. The other months were the driest, especially the month of November. On the other hand, in the Sertão Pernambucano (Figure 3d) and Sertão do São Francisco (Figure 3e) mesoregions, the rainy season occurred from December to May, with maximum rainfall in February and March, and the dry period occurred in the other months (June to November), where the month of September registered the lowest rainfall. Thus, it is possible to evidence the variation of the average monthly rainfall between the different mesoregions of the state of Pernambuco.

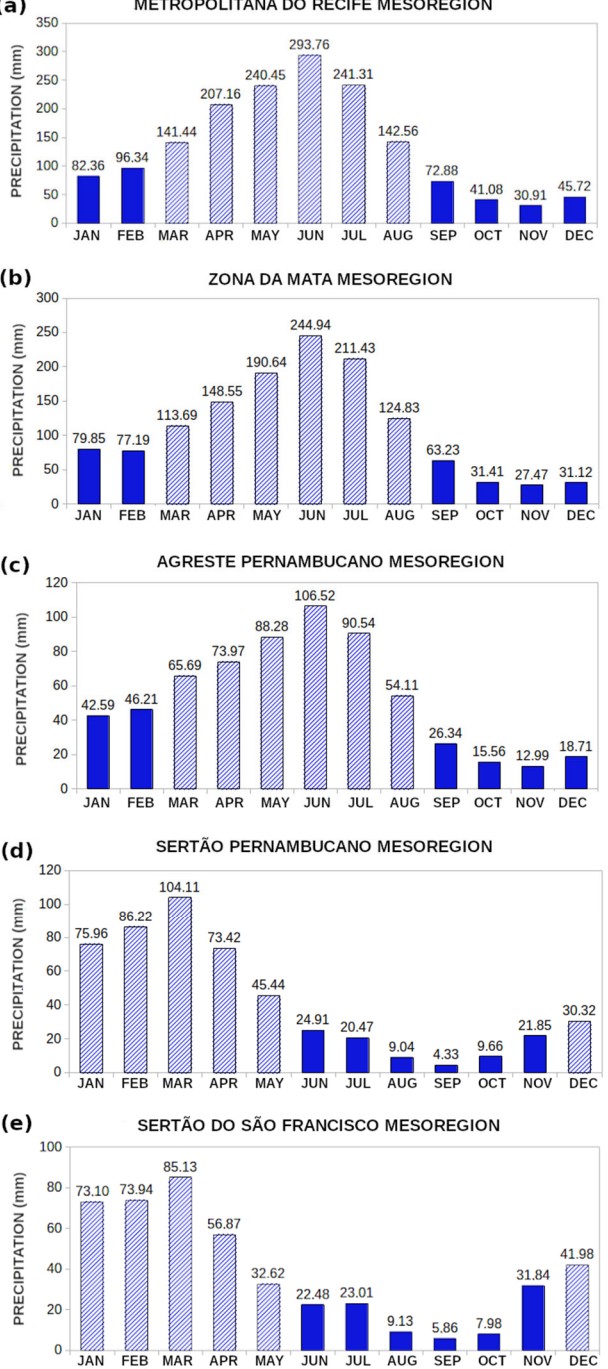

**Figure 3.** Monthly rainfall in the mesoregions (**a**) Metropolitana do Recife, (**b**) Zona da Mata, (**c**) Agreste Pernambucano, (**d**) Sertão Pernambucano, and (**e**) Sertão do São Francisco, with emphasis on the rainy months in the period from 1990 to 2020.

### 3.1. The Precipitation Concentration Index

Figure 4 shows the relative frequency of the *PCI* in the mesoregions of Pernambuco from 1990 to 2020. The results show that, during the analyzed period, none of the state's mesoregions presented a uniform rainfall distribution (*PCI* ≤ 10). Figure 5 shows the average annual *PCI* for the entire state of Pernambuco over the 31 years analyzed. The results show that the values of this index ranged between 13.0 and 31.0, with the highest values in the Sertão Pernambucano and Sertão do São Francisco mesoregions and lower values in the Metropolitana do Recife and Zona da Mata mesoregions.

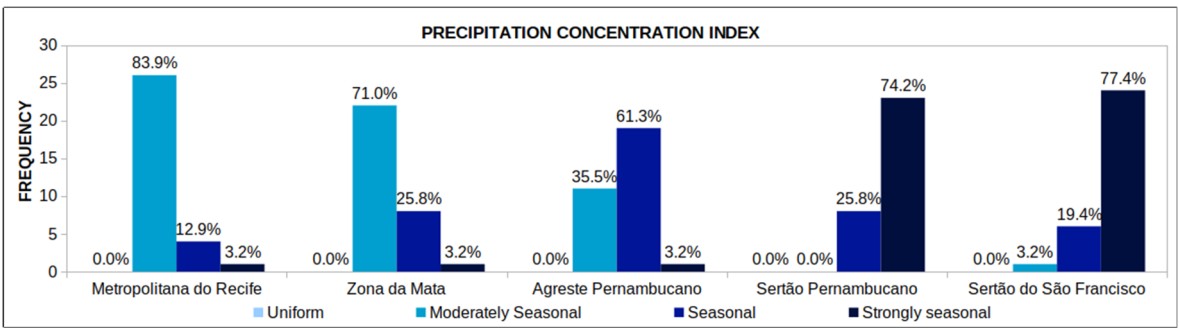

**Figure 4.** Relative frequency of the *PCI* in the mesoregions of the state of Pernambuco from 1990 to 2020.

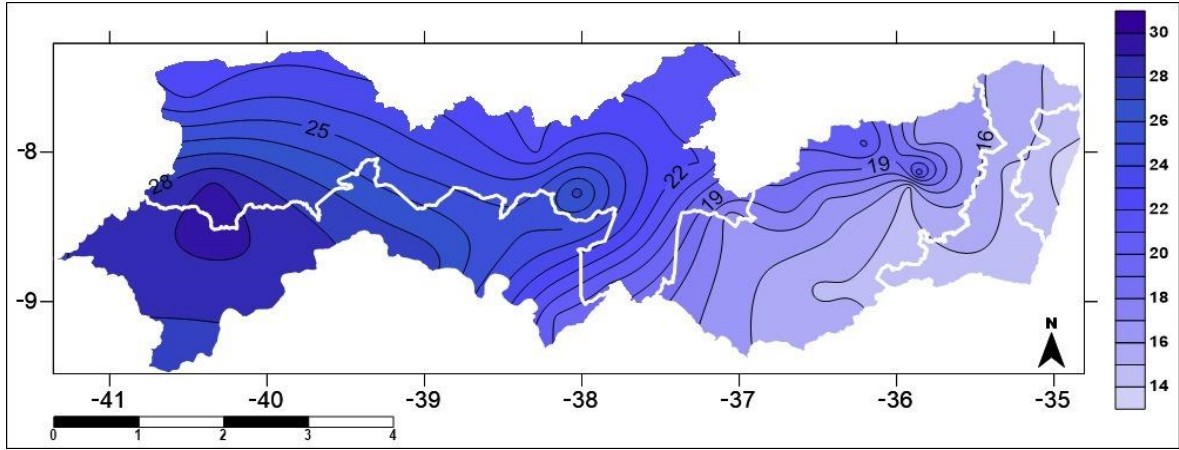

**Figure 5.** Spatial distribution of the annual average *PCI* for the state of Pernambuco from 1990 to 2020.

The annual average *PCI* for the Metropolitana do Recife (Figure 6a) and Zona da Mata (Figure 6b) mesoregions were, respectively, 14.51 and 15.05, indicating that rainfall in these mesoregions had a moderately seasonal distribution. Considering the annual *PCI*, in the Metropolitana do Recife mesoregion, the values of this index varied between 10.44 and 20.44, while in the Zona da Mata mesoregion, the values varied between 11.31 and 21.99. For Agreste Pernambucano (Figure 6c), the average annual *PCI* was 17.14, classifying the rainfall distribution in this mesoregion as seasonal. Concerning the annual *PCI*, this index's maximum and minimum values were, respectively, 23.05 in 1998 and 12.27 in 2000. The highest values of the *PCI* were observed in the mesoregions of Sertão Pernambucano (Figure 6d) and Sertão do São Francisco (Figure 6e), where the rainfall distribution was strongly seasonal, indicating a relative frequency of 74.2% in Sertão Pernambucano and 77.4% in Sertão do São Francisco, within the analyzed period (Figure 5).

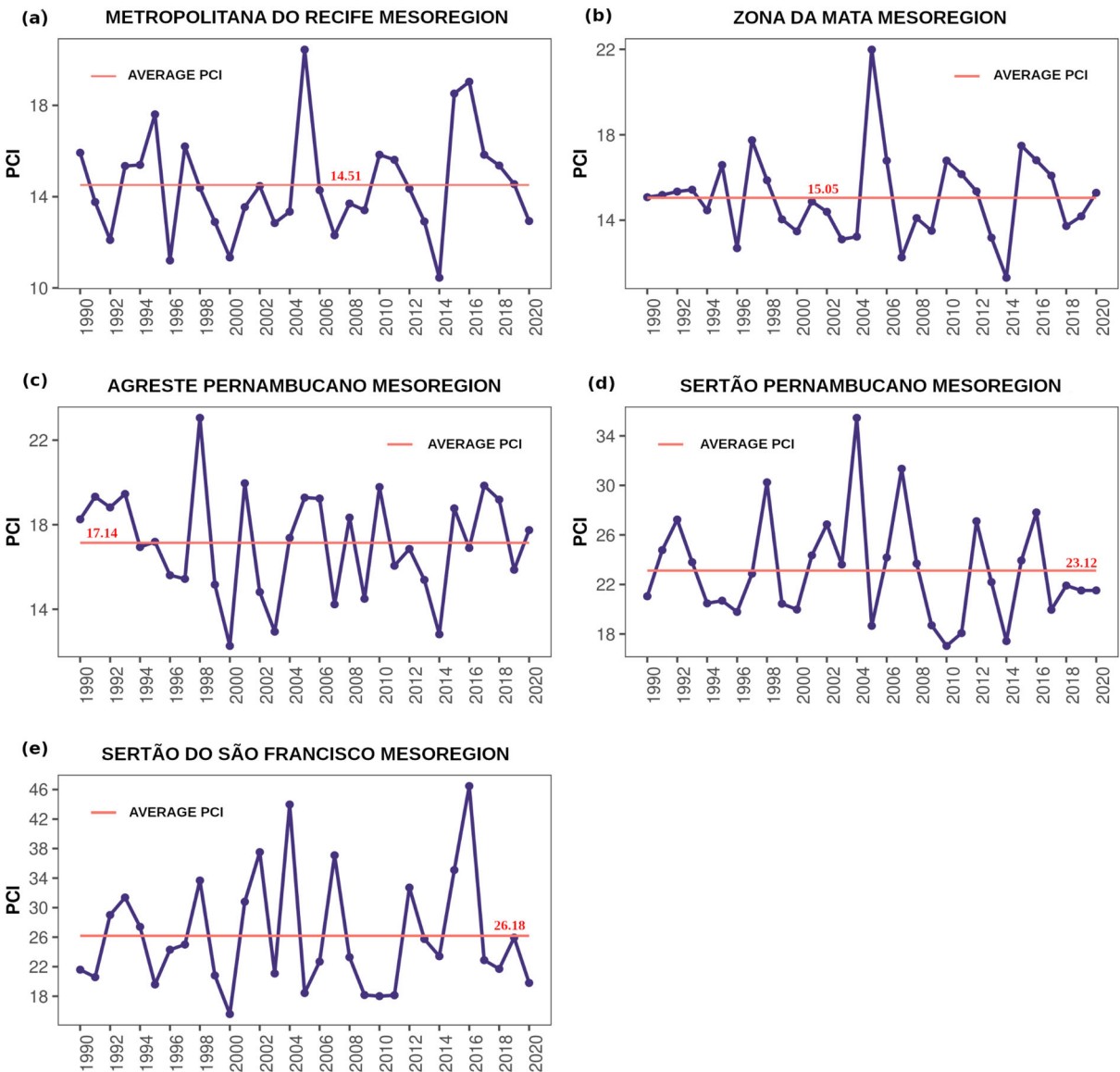

**Figure 6.** The Precipitation Concentration Index for the period from 1990 to 2020 in the mesoregions (**a**) Metropolitan of Recife, (**b**) Zona da Mata, (**c**) Agreste Pernambucano, (**d**) Sertão Pernambucano and (**e**) Sertão do São Francisco.

### *3.2. The Standardized Precipitation Index*

Figure 7a presents the distribution of the average *SPI* on the quarterly scale for the rainfall stations used, where the *SPI* indicated intense droughts in the state in 1993, when peaks of extreme drought occurred, from 1998 until the end of 1999, in 2012, between 2016 and 2017 and in 2018, with months of severe drought. It was also verified that between 2000 and 2011, there were several rainy events, and 2004 stood out for extreme rains, and the period from 2009 to 2011 for having a long rainy period, where moderate rains were predominant.

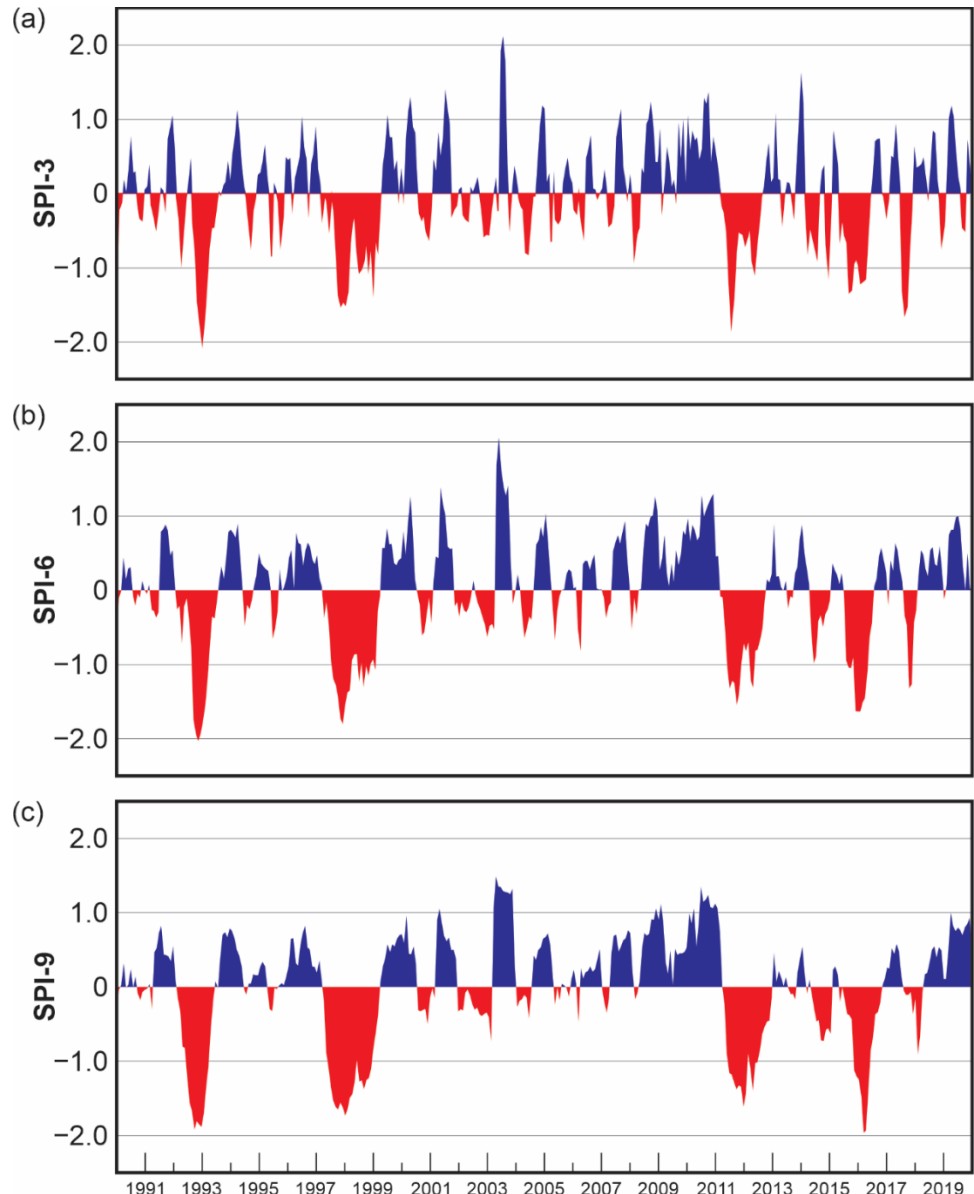

**Figure 7.** Distribution of the average *SPI* on the quarterly scale for the rainfall stations used in this study (**a**) *SPI-3*, (**b**) *SPI-6*, (**c**) *SPI-9*.

On the semi-annual scale (Figure 7b), it is observed that the droughts that occurred between 1998 and 1999 and between 2016 and 2017, which were indicated by the *SPI-3*, were more expressive, indicating that the rainfall that occurred in the period was not sufficient to meet water needs on a larger scale. About rainy events, it is observed that light rains became predominant, especially in the 1990s and after the 2010s. Finally, on the 9-month scale (Figure 7c), there were no significant changes concerning the six-month scale since the dry and rainy events identified in the *SPI-6* remained. The main highlight on this scale was the intensification of the drought between 2012 and 2013 and the rains in 2004, which went from extreme to severe rain category.

Analyzing the results of the *SPI-12* for each of the mesoregions of the state, we have that in the Metropolitan mesoregion of Recife (Figure 8a), there was a higher frequency of low and moderate rains. However, severe rains occurred at a lower frequency. Notably, most rainfall events were concentrated mainly in the first half of the 1990s, except for the dry period between 1993–1994, when intense droughts affected the entire state. The most significant drought period that hit this mesoregion with great intensity occurred between 1998 and 2000, where 1998 presented several consecutive months of extreme drought.

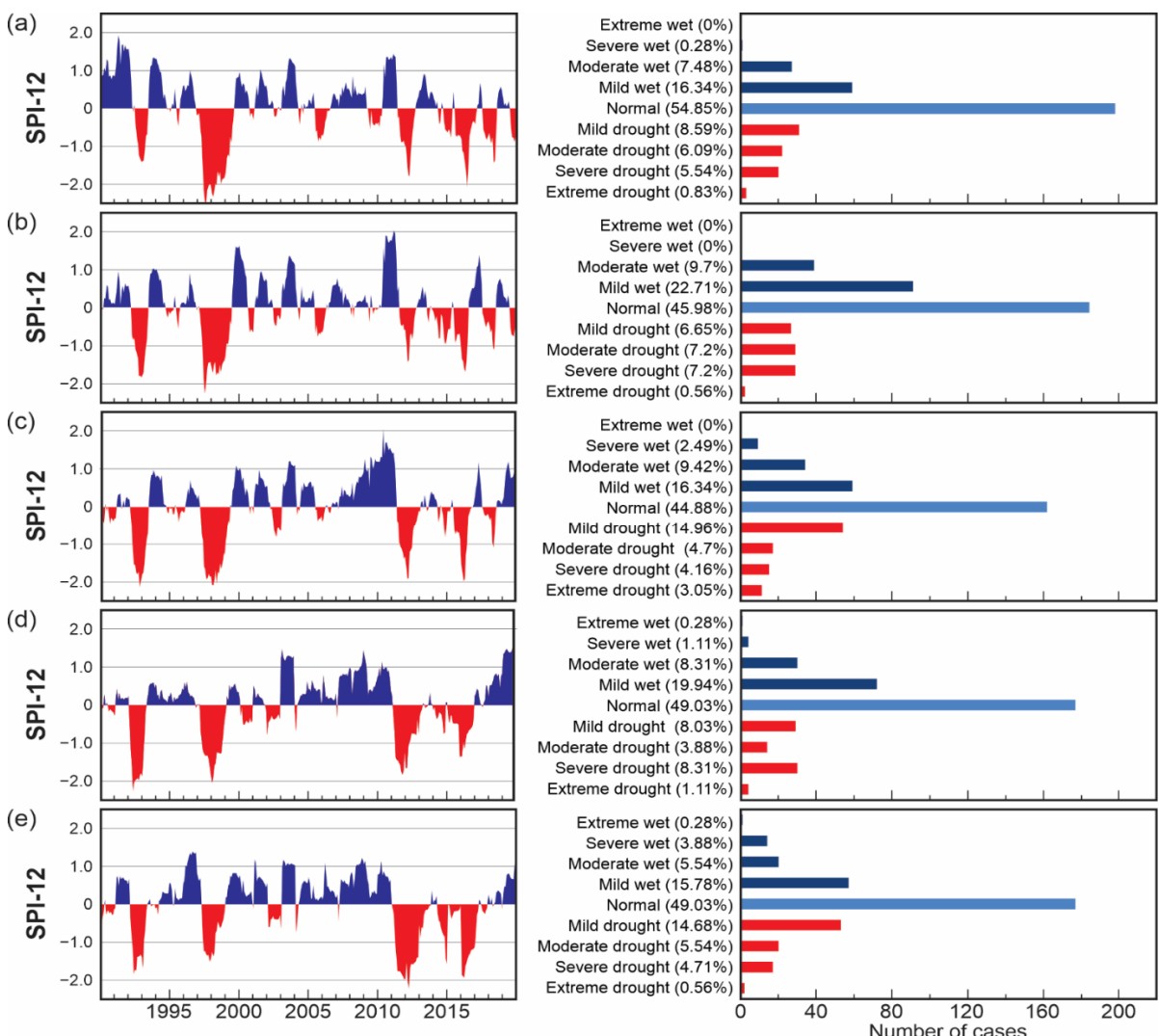

**Figure 8.** *SPI-12* time series and distribution of the frequency of events by the *SPI-12* category for the (**a**) Metropolitana do Recife; (**b**) Zona da Mata; (**c**) Agreste Pernambucano; (**d**) Sertão Pernambucano and (**e**) Sertão do São Francisco mesoregions.

From the 2000s onwards, it is observed that weak and moderate rains were predominant, with no severe and extreme rains having been recorded. In the Zona da Mata (Figure 8b), it is observed that unlike the Metropolitana do Recife mesoregion, the 1990s were characterized by less intense rainfall events and that from the 2000s onwards, there was a greater predominance of rains in moderate and severe categories, with peaks of extreme rainfall in 2012. Regarding drought events, it appears that there were intense droughts in the 1990s and that, in both mesoregions, from the 2010s onwards, droughts passed to be predominant. The highlight of these two mesoregions was the year 2020, which, unlike what was observed in the other mesoregions, presented episodes of mild drought.

In Agreste Pernambucano (Figure 8c), it is observed that weak and moderate rains were predominant, mainly in the 2000s. In this mesoregion, the highlight is the rainy period between 2009 and 2012. The *SPI* time series recorded peaks of severe and extreme rains, the most prolonged rainy period observed in all mesoregions. Concerning drought events, weak and severe droughts were predominant, where droughts occurring between 1993–1994 stand out, 1998–2000 when there were peaks of extreme droughts, 2012–2013, and 2016–2017.

In the Sertão Pernambucano mesoregion (Figure 8d), there were intense droughts in 1993–1994 and 1998–1999, reaching peaks of extreme drought and a higher frequency of severe droughts concerning the droughts that occurred in the 2010s. The opposite happened in the Sertão do São Francisco mesoregion (Figure 8e). The droughts in the 2010s

(2012–2014, 2015 and 2017–2018) were more intense than those in the 1990s, mainly showing dryness in the extreme and severe categories. Regarding rainfall, it is observed that in the Sertão Pernambucano mesoregion, there was a predominance of low and moderate rains, mainly in the period from 2004 to 2011 and that, after a period of significant drought, from 2019 onwards, the events became positive again, with emphasis on 2020, which presented moderate and severe rains. The results also showed that Sertão do São Francisco mesoregion was characterized by not having registered extreme and severe rains, highlighting the rainy events that occurred in 1996–1998, most of the 2000s, 2011 and 2020.

### 3.3. *The Precipitation Concentration Degree and the Precipitation Concentration Period*

Figure 9 illustrates the spatial distribution of *PCD* values for the entire state of Pernambuco, in which considerable variations in precipitation concentration can be observed, reflecting the tremendous climatic variability of the state of Pernambuco. It can be seen from the figure that the area located between the microregions of Vale do Ipanema and Sertão do Moxotó has the lowest *PCD* value, indicating that rainfall is more homogeneous throughout the year in this location. *PCD* values are also low, reflecting better rainfall distributed throughout the year in the Metropolitana do Recife, Zona da Mata, and Agreste mesoregions. As we move towards the west region, it is observed that the values of this index are increasing, reaching the highest value in the extreme west of the state, indicating that the rains in these locations are poorly distributed and concentrated in a short period of the year. For the *PCP* (Figure 10), it is observed that the mesoregions Metropolitana do Recife, Zona da Mata and part of Agreste have rains concentrated in May, and, as it moves towards the west of the state, this period changes for April, in large part of the Agreste region, for March in the Sertão do São Francisco mesoregion and February in the Sertão Pernambucano.

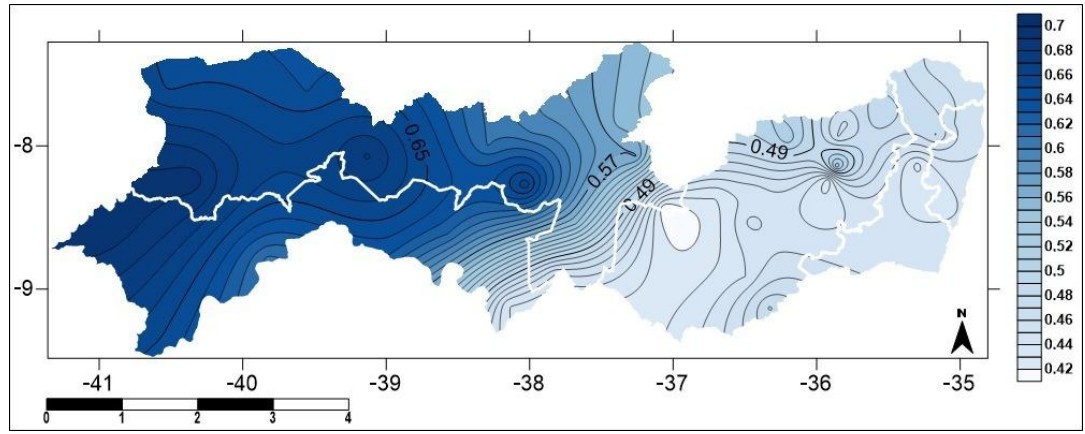

**Figure 9.** *PCD* for the state of Pernambuco for the period from 1990 to 2020.

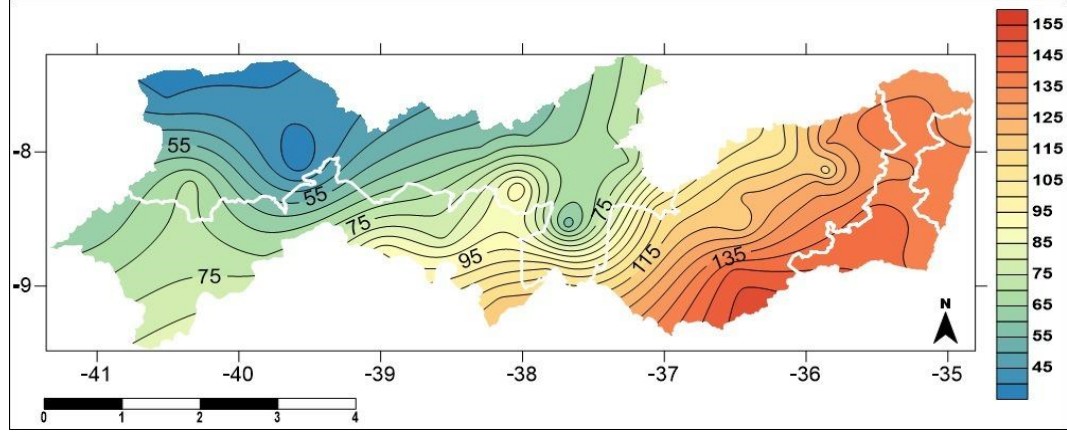

**Figure 10.** The *PCP* for the state of Pernambuco for the period from 1990 to 2020.

### 3.4. The Rainfall Anomaly Index

Figure 11 describes the results obtained for the *RAI* in all the mesoregions of the state of Pernambuco. In the Metropolitana do Recife mesoregion (Figure 11a), 1992 is classified as highly wet, and 1994, 2004 and 2011 as high wet. Regarding the negative anomalies, 1998 and 1999 were extremely dry. In Zona da Mata (Figure 11b), dry years were more frequent than wet years or years of normal variability, emphasizing 1993 and 1998 as extremely dry and the years 2000 and 2011 as being characterized as extremely wet. In addition, it is identified that from 2015, in both mesoregions, there was a decrease in the years classified as rainy and an increase in the years of drought.

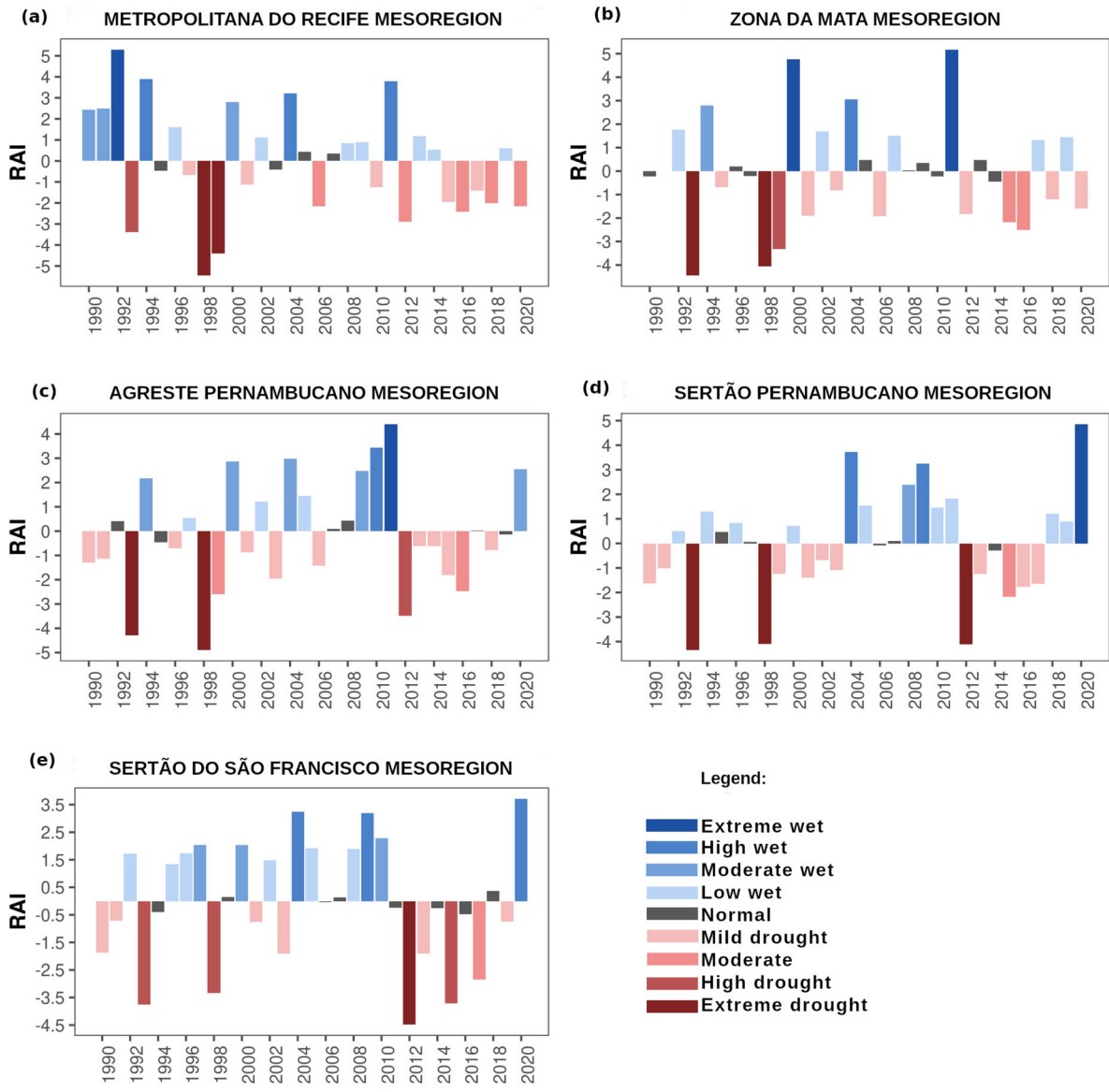

**Figure 11.** The Annual Rainfall Anomaly Index for the period from 1990 to 2020 in the mesoregions (**a**) Metropolitan of Recife, (**b**) Zona da Mata, (**c**) Agreste Pernambucano, (**d**) Sertão Pernambucano and (**e**) Sertão do São Francisco.

In the Agreste mesoregion (Figure 11c), the years 1993 and 1998 stood out for, in addition to being marked by the lowest rainfall, they presented the prominent negative peaks in the *RAI*, being classified as extreme drought, while the years 2010 and 2011 were classified, respectively, as high wet and extreme wet. In addition, it is noted that there was a predominance of negative over positive anomalies in this mesoregion.

For Sertão Pernambucano (Figure 11d), we have that until 2003, except for 1993 and 1998, which were extremely dry years, the *RAI* values oscillated between −1.6 and 1.3, reflecting years of mild drought, with variability normal or low wet. From 2004 onwards, there were wet, normal, and dry years, highlighting 2012, which was an extremely dry year, and 2020, which was extremely humid. In the Sertão do São Francisco mesoregion (Figure 11e), it is observed that no year was classified as extremely humid, with 2020 having the highest value of this index and being classified as having high humidity. Regarding the negative anomalies, it is noteworthy that the years classified as extremely dry or high drought occurred more frequently after the 2010s. However, this index recorded years of high drought in 1993 and 1998.

### 3.5. Trend Analysis in the Annual Rainfall and Climate Indices Time Series

Table 6 shows the statistics from the Mann–Kendall and Sen's slope tests for the average annual rainfall data. Analyzing the table, none of the state's mesoregions demonstrated trends in this variable since their *p*-values were larger than the level of significance specified for this study. Sen's test revealed that except for the Sertão do São Francisco mesoregion, all other mesoregions had a magnitude in the historical series, with negative magnitudes in the Metropolitana do Recife and Zona da Mata mesoregions and positive magnitudes in the Agreste and Sertão Pernambucano mesoregions. However, because the null hypothesis was not rejected in any mesoregion, none of these magnitudes are significant.

**Table 6.** Statistics from Mann–Kendall and Sen's slope tests for mean annual rainfall from 1990 to 2020.

| Mesoregion | Z | Significance | Sen's Slope |
|---|---|---|---|
| Metropolitana do Recife | −1.7676 | 0.07712 | −12.7317 |
| Zona da Mata | −0.16996 | 0.865 | −1.1733 |
| Agreste Pernambucano | 1.2237 | 0.221 | 4.69134 |
| Sertão Pernambucano | 0.8498 | 0.3954 | 3.8586 |
| Sertão do São Francisco | 0.067986 | 0.9458 | 0.2108 |

Table 7 presents the statistics of the Mann–Kendall and Sen's slope tests for the *PCI*, the *SPI-12*, the *PCD*, the *PCP*, and the *RAI*. Considering the *PCI* results, all mesoregions presented *p*-values greater than the chosen significance ($\alpha = 0.05$), thus demonstrating no tendency for this index in any mesoregions. The table also shows that Sen's test indicated values very close to zero, demonstrating the low magnitude of the *PCI* historical series in the five analyzed mesoregions. For the *SPI-12*, the Mann–Kendall test showed no trends in the time series of this index in the Zona da Mata, and Sertão do São Francisco mesoregions. However, in the Metropolitana do Recife, Agreste Pernambucano, and Sertão Pernambucano mesoregions, *p*-values are below the specified significance, thus obtaining evidence to reject the null hypothesis, that is, there is a tendency present in the *SPI-12* series of these mesoregions. However, when analyzing the results of Sen's test, it appears that this test found values very close to zero in these locations, so the magnitude of these trends is not significant. Finally, when analyzing the results of the Mann–Kendall test for the *PCD*, the *PCP*, and the *RAI*, it is observed that all mesoregions of the state presented *p*-values considerably higher than the significance of $\alpha = 0.05$, making it impossible to reject the null hypothesis that there is no trend in the analyzed series of these three indices. The second test's values are very close to zero, demonstrating that the magnitudes found are insignificant.

**Table 7.** Mann–Kendall and Sen's slope test statistics for the indices used in this work.

| Index | Mesoregion | Z | Significance | Sen's Slope |
|---|---|---|---|---|
| PCI | Metropolitana do Recife | 0.37 | 0.71 | 0.01 |
| | Zona da Mata | −0.14 | 0.89 | −0.01 |
| | Agreste Pernambucano | −0.58 | 0.56 | 0.02 |
| | Sertão Pernambucano | −0.88 | 0.38 | −0.08 |
| | Sertão do São Francisco | 0.07 | 0.95 | 0.02 |
| SPI-12 | Metropolitana do Recife | −5.521 | <0.005 | −0.002465 |
| | Zona da Mata | −1.3771 | 0.1685 | −0.00056 |
| | Agreste Pernambucano | 2.6053 | 0.0092 | 0.00103 |
| | Sertão Pernambucano | 3.3517 | 0.0008 | 0.0013 |
| | Sertão do São Francisco | −1.2898 | 0.1971 | −0.0005 |
| PCD | Metropolitana do Recife | 0.64586 | 0.5184 | 0.0014 |
| | Zona da Mata | −0.67986 | 0.4966 | −0.0019 |
| | Agreste Pernambucano | −0.27194 | 0.7857 | −0.00058 |
| | Sertão Pernambucano | 0 | 1.00 | −0.0000038 |
| | Sertão do São Francisco | 0.10198 | 0.9188 | 0.0002 |
| PCP | Metropolitana do Recife | −0.88381 | 0.3768 | −0.4680 |
| | Zona da Mata | −0.81503 | 0.4146 | −0.3199 |
| | Agreste Pernambucano | −0.16996 | 0.8650 | −0.1791 |
| | Sertão Pernambucano | −0.33993 | 0.7339 | −0.1599 |
| | Sertão do São Francisco | −0.71385 | 0.4753 | −0.4371 |
| RAI | Metropolitana do Recife | −1.6656 | 0.0958 | −0.0897 |
| | Zona da Mata | −0.27194 | 0.7857 | −0.0118 |
| | Agreste Pernambucano | 0.86694 | 0.3860 | 0.03922 |
| | Sertão Pernambucano | 0.88381 | 0.3768 | 0.0487 |
| | Sertão do São Francisco | 0 | 1 | 0.00014 |

## 4. Discussion

Rainfall is one of the essential elements in tropical regions since it, directly and indirectly, influences human activities, the economy, and the environment. In this sense, accurate knowledge of the characteristics of this variable is of fundamental importance for planning and monitoring the impacts caused by excess or scarcity of rainfall [10,78]. Considering the spatial distribution of average annual rainfall in the state of Pernambuco from 1990 to 2020, it was found that, in quantitative terms, the highest rainfall rates were concentrated in the Metropolitana do Recife, and Zona da Mata mesoregions, followed by the Agreste mesoregions of Sertão Pernambucano and Sertão do São Francisco. This result was already expected since, in the mesoregions Metropolitana do Recife and Zona da Mata, due to the proximity to the ocean, there is the influence of meteorological systems that act in the eastern part of the state, providing annual rainfall varying up to 2200 mm [79].

These results are similar to other studies carried out in the state. For example, [62], who observed rainfall of more than 1000 mm/year on the coast and Zona da Mata, accumulated between 600 and 800 mm/year in the Agreste Pernambucano mesoregion and rainfall of between 400 and 800 mm/year in the Sertão and Sertão do São Francisco mesoregions. Andrade et al. (2018), [80] observed that the average rainfall is 944.44 mm/year in the Agreste Meridional.

The interannual variability of precipitation in the state can be explained by several factors, such as the local geography and the action of atmospheric and oceanic phenomena, from the teleconnections. We can highlight the ENSO and the SST Anomaly Gradient in the Tropical Atlantic. These settings determine the performance and intensity of the various meteorological systems in this region. Positive ENSO (El Niño) phases and the

SST Anomaly Gradient in the Tropical Atlantic (presence of positive anomalies in the North Atlantic and negative anomalies in the South Atlantic) cause precipitation to decrease in the NEB region [36,81].

It is also worth noting that, in the state of Pernambuco, most years with below-average rainfall were associated with the El Niño phenomenon and/or the positive phase of the SST Anomaly Gradient in the Tropical Atlantic [82]. According to Pereira et al. (2017), [83] in the driest years, several municipalities in the state, especially those located in the mesoregions of Agreste, Sertão, and Sertão do São Francisco, declared a state of public calamity and/or emergency, where rural areas were the ones that suffered the greatest impacts, with great losses in agriculture and livestock.

Furthermore, our results agree with those found by Santos et al. (2014) [84], Alves et al. (2016) [85], Mendonça (2016) [86], Pereira et al. (2017) [83], Silva et al. (2018) [87], and Costa et al. (2021) [82], who in their studies in the state of Pernambuco proved that the rainy season in the mesoregions of Metropolitana do Recife, Zona da Mata, and Agreste Pernambucano occurs mainly in June and July and that in the mesoregions of Sertão Pernambucano and Sertão do São Francisco, the rains are predominant in February and March. According to Araújo et al. (2009) [88], the month of March is the most significant in terms of rainfall in the state's interior due to the ITCZ's activity, which reaches its maximum in that month when it becomes prominent in the region.

Based on the results presented, it was found that the annual rainfall in the state of Pernambuco is not uniformly distributed. According to Souza et al. (2021) [89], this result may be related to the fact that the spatial variability of the climate considerably influences the average *PCI* of a given location. In their studies, Sangüesa et al. (2018) [52] found that the annual *PCI* tends to decrease by up to 30% when moving from an arid/semi-arid zone to a humid/sub-humid zone. Thus, the state's rainfall distribution can be moderately seasonal to strongly seasonal, where the Sertão Pernambucano and Sertão do São Francisco mesoregions recorded the highest index values. These values suggest that, in these locations, much of the annual rainfall is concentrated in short periods, which can cause droughts, which have a considerable impact on water resources and water supply, or floods [53].

In the analyses carried out by the *SPI*, it was found that the rainy events occurred mainly between the years 2000 and 2011 and that the years 1993, 1998–1999, 2012, 2016–2017 and 2018 were marked by intense droughts in agreement with the results found by Farias et al. (2014) [61], Guedes et al. (2016) [90], Brito et al. (2018) [27], Barros et al. (2021) [91] and Santos et al. (2021) [92] in their studies in the NEB region using the *SPI*. According to De Nys et al. (2016) [93] and Lima and Magalhães (2019) [94], historically, the NEB region has been plagued by major droughts since the 16th century, especially the droughts that occurred in 1877–1879, 1992–1993, 1997–1998, 2001–2002, 2005, 2007, 2010 and 2012–2017, which were responsible for significant economic losses, livestock deaths, a decline in agricultural production, and consequently, migrations to other regions of the country.

The *PCD* results suggest that the lowest values of this index are observed in the state's central region. This happens because the rainfall in this region is influenced by the actions of systems that act both in the west, such as the ITCZ and the High-Level Cyclonic Vortices, and in the east of the state [5,95,96]. The state's western region presented the highest values of this index, suggesting the influence of meteorological systems, when active, favoring the occurrence of rain in a single period of the year [97]. Similar results were found by Araújo (2013) [98] and Silva and Lucio (2015) [99], who, when using the *PCP* and the *PCD* to characterize the temporal variability of rainfall in Rio do Grande do Norte, the state of the NEB region, concluded that the eastern region of the state presents a better distribution of rainfall throughout the year. At the same time, the west tends to show irregular precipitation and is concentrated in a few months of the year.

The use of the *RAI* made it possible to evaluate and classify the degree of severity of the dry and rainy periods in the state of Pernambuco, where it indicated that, in most of the state, the years 1993, 1998 and 2012 were characterized by being extremely dry. The

year 1998 was marked by extreme drought throughout northeastern Brazil where Pernambuco State, as a whole, had a water deficit, affecting the development of economic activities, water reserves, and human supply [100]. In addition, the application of this index made it possible to identify that, in the Sertão do São Francisco mesoregion, extreme negative anomalies were more frequent after 2010, in agreement with the studies by Brito et al. (2018) [27]. They analyzed the drought events that occurred in the semi-arid region of the NEB region and found that between 2011 and 2016, the most severe and prolonged drought since 1981 occurred, with significant impacts on the population and economic activity. Diniz et al. (2020) [101] found this same behavior in rainfall in Cariri Occidental Paraibano, a microregion located in the NEB region, when examining the climate variability of the microregion from 1999 to 2019, finding a greater record of years classified as extremely dry, high drought, and moderate drought after 2010. The results of this study also agree with those found by Costa and Da Silva (2017) [102]. When analyzing the space-time variation of rainfall in the state of Ceará, NEB region, from 1973 to 2010, they observed a higher frequency of dry years than wet years, mainly after 1991, indicating a possible change in the local rainfall pattern.

Analyzing the statistics of the Mann–Kendall and Sen slope tests described in Tables 6 and 7, we observed that none of the mesoregions of the state of Pernambuco showed trends in mean annual precipitation during the analyzed period. Our results are similar to those of Salviano et al. (2016) [103], who analyzed the temporal trends of precipitation and the average temperature in Brazil from 1961 to 2011. Barbosa et al. (2016) [104], Alcântara et al. (2019) [105], Verçosa et al. (2019) [106] and Bezerra et al. (2021) [107], in different areas of the state of Pernambuco, Gonçalves and Back (2018) [108], in the southern region of Brazil, Thielen et al. (2020) [109], in the Upper Paraguai basin, and Cabral Júnior and Lucena (2020) [110], in the state of Rio Grande do Norte, in which no trends were identified in the time series of precipitation. Of all the calculated indices, only the *SPI-12* showed trends in some mesoregions. However, Sen's test indicated that the magnitudes of the trends found are not significant. In studies also carried out in the state of Pernambuco, Silva et al. (2020) [111] and Barros et al. (2021) [91] used the Mann–Kendall test and found trends in the *SPI* series on the 12-month scale, indicating that in some locations in the state, there was an increase and a decrease in dry and wet periods in recent years.

## 5. Conclusions

The analysis and understanding of the distribution of precipitation in a given location are of fundamental importance for society, given that under extreme weather conditions, such as droughts and floods, various social, environmental, political, and economic aspects can be affected. Thus, it is necessary to understand in detail the historical behavior of precipitation to create measures to prevent, mitigate, and adapt to the impacts caused by this phenomenon.

This study analyzed rainfall's spatial and temporal variability in the State of Pernambuco. As a result, it was possible to identify several aspects of this variable using five climate indices: the rainfall regime, the dry and rainy seasons; the distribution and concentration of rainfall; the occurrence of extremes; and temporal trends. Thus, based on the results obtained, it can be concluded that: (i) annual rainfall does not have a uniform distribution, being classified as moderately seasonal in the Metropolitana do Recife and Zona da Mata mesoregions, seasonal in the Agreste Pernambucano mesoregion, and strongly seasonal in the Sertão Pernambucano and Sertão do São Francisco mesoregions; (ii) according to the *SPI*, events classified as normal were the most frequent in all mesoregions and, in addition, it was possible to identify a greater number of rain events as opposed to drought events, although the intensity and duration of droughts were higher throughout the state, with droughts in the extreme category for several consecutive months in some mesoregions. In addition, drought events with greater spatial coverage, mainly in the categories of severe and extreme droughts, occurred in almost the entire state in the 1990s, except for the Sertão do São Francisco mesoregion, where they predominated in the 2010s;

(iii) the *PCD* and *PCP* indices indicated that the east of the state has more distributed rainfall throughout the year, with the rains concentrated in May; (iv) Pernambuco has large annual fluctuations in rainfall—in the last decade, in most of the state, there was a greater occurrence of negative anomalies than positive anomalies, indicating a possible modification in rainfall patterns; (v) there are no trends in any of the mesoregions in the time series of annual rainfall and the *PCI*, *PCD*, *PCP* and *RAI* indices, while the *SPI-12* showed trends in the mesoregion Metropolitana do Recife.

It is expected that the results obtained in this work will be helpful for a better understanding of the rainfall events that occurred in the state of Pernambuco during the analyzed period. Therefore, it is recommended that other locations in the NEB region be included in future studies or even the use of different indices that involve other meteorological variables.

**Author Contributions:** T.R.B.F.S. and C.A.C.d.S. contributed to the study conception and design. Material preparation, data collection, and analysis were performed by T.R.B.F.S. and D.J.F.S. The first draft of the manuscript was written by T.R.B.F.S., C.A.C.d.S., D.J.F.S., C.A.G.S., R.M.d.S. and J.I.B.d.B. and all authors commented on previous versions of the manuscript. All authors read and approved the final manuscript. All authors have read and agreed to the published version of the manuscript.

**Funding:** This work was supported by the Conselho Nacional de Desenvolvimento Científico e Tecnológico (CNPq) for providing the Research Productivity Grant (Process No. 304493/2019-8).

**Institutional Review Board Statement:** No individual personal data are included in this study.

**Informed Consent Statement:** All authors agree to the publication of the study.

**Data Availability Statement:** The datasets analyzed during the current study are available on the Pernambuco Water and Climate Agency (APAC) website (http://old.apac.pe.gov.br/meteorologia/monitoramento-pluvio.php (accessed on 15 January 2021)) and the National Institute of Meteorology (Inmet) (https://bdmep.inmet.gov.br/ (accessed on 25 February 2021)).

**Acknowledgments:** The second author acknowledge the Conselho Nacional de Desenvolvimento Científico e Tecnológico (CNPq).

**Conflicts of Interest:** The authors declare no competing interests.

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
