# Peer review of "Climate Indices-Based Analysis of Rainfall Spatiotemporal Variability in Pernambuco State, Brazil"

_water, doi:10.3390/w14142190_

Round 1

Reviewer 1 Report

In the paper absent the mention of quality control of input precipitation data. I suggest adding this to the chapter "2. Material and Methods"

Author Response

Answer to Reviewer 1

Dear Reviewer,

We would like to thank you for reading, reviewing, and evaluating this work, which will undoubtedly contribute to improvements in the quality of our text. Thus, we present below the answers to the questions raised and some suggestions.

Comments and Suggestions for Authors

1) In the paper absent the mention of quality control of input precipitation data. I suggest adding this to the chapter "2. Material and Methods"

Answer: Following the reviewer's suggestion, we added in Chapter 2 more information on the quality control of precipitation data.

Author Response

Answer to Reviewer 2

Dear Reviewer,

We would like to thank you for reading, reviewing, and evaluating this work, which will undoubtedly contribute to improvements in the quality of our text. Thus, we present below the answers to the questions raised and some suggestions.

Comments and Suggestions for Authors

  1. Abstract:
    The Abstract needs to re-clarified. The authors said the PCI and PCD results indicate a seasonality in local rainfall, but how and why? At least some detailed numbers should be concluded to verify this result. Besides, “the SPI revealed that the most and extreme droughts....”. Again, how the SPI reveal this result? The exact values of SPI, or at least the spatially-averaged SPI values during the 1990-2020 period should be added to explain this result. Finally, the authors said “a higher frequency of negative anomalies than positive anomalies in the last decade”. Then what is the high frequency and paired low frequency?

Answer: As suggested by the reviewer, the abstract was rewritten so that most of the questions could be answered.

  1. Introduction:

The authors used three paragraphs to describe general features of climate change and its impacts on extreme events (The first three paragraphs). However, more details concerning about the exact occurred extreme rainfall and droughts in recent decades are neglected. Specifically, 1. More potential mechanisms about climate change impacts on extreme precipitation events and drought episodes should be added; e.g., climate warming increases the water holding capacity, enabling more extreme precipitation and subsequently more frequent droughts (Lehner et al., 2017; Gu et al., 2020; 2021). 2. The authors should at least employ some examples, the recent occurred droughts and extreme precipitation, to strengthen the impression of their impacts on the ecological
and social sectors. 3. In describing conditions in Brazil, again, examples about the already occurred extreme precipitation and drought events should at least be briefly introduced, or how previous studies focus on and investigate extreme events in Brazil? Moreover, I still think the authors should dig further about the objective and creativity of this study. “Analyze the spatial and temporal variability of rainfall in Pernambuco and the occurrence of extreme events” seems too general.

Answer: The introduction of the work was completely rewritten, and new information was inserted throughout the text to answer questions and meet the reviewer's suggestions. Concerning examples of extreme rainfall and drought events in Brazil, the most significant extreme events that occurred in the study area in the last 30 years were added in the penultimate paragraph. As for the objectives of the work, new arguments were added to make it clearer. 

3 Results
There are too many figures in this article, most of them are not that important and necessary and should be displayed in supplementary materials (SM). Further, some of them should be aggregated into a bigger figure. For example, Figure 2 and 3 can be aggregated and put in the SM, I donot think it is meaningfulness to simply figure the annual precipitation. SPI3-SPI12 figure should also be aggregated, then the authors should analyze which periods were dry and which periods were wet. In fact, 1990-2020  is a quite long period, and there should be some tendency in SPI series. The authors can investigate the slope of SPI and the associated seasonality. In this way, we can know
whether drought conditions become more severe in recent decades than 1990s, and whether spring was wetter or drier than summer or winter (or, droughts tend to occur in what season?) Moreover, too many SPI figures, please aggregate and move redundant SPI figures to SM.

Answer: As suggested by the reviewer, the number of figures in this article was reduced, most of which were grouped. As for the presentation of annual rainfall in the text, this was done to show significant differences in rainfall from one mesoregion to another over the analyzed period. Regarding the investigation of the slope of the SPI and the associated seasonality, we will develop another study, which will be part of doctoral research, where we will address the same indices used in this work, and the slope of the indices is an integral part of this study. 

Reviewer 3 Report

The paper investigates rainfall spatiotemporal variability in Pernambuco State based on several climatic indices. The paper is well written but devoid of originality.

Below I outline my suggestions for further improvement:

Major comments:

1.  What is the novelty of the work reported in this manuscript? The methods used in the study are all standard procedures used for similar kinds of studies in different regions of the world. 

2. The assessment of extreme rainfall with the indices may not provide the actual representation. The authors need to add indices such as annual maximum rainfall of 1-day (and or 5-day) duration. This would provide further information about rainfall severity.

3. Some similar indices are used (such as SPI and RAI). Which one is more appropriate in the study area? This should add some innovation to the study.

Minor comments:

1. Title should include the country name Brazil

2. Page 1, last line: Authors should include more citations in the statement including recent studies such as (Das et al. 2022) and IPCC reports

3. What is the concrete outcome from the research- vulnerable to extreme rainfall/ drought?

References:

Das S, Kamruzzaman M, Islam ARMT (2022) Assessment of characteristic changes of regional estimation of extreme rainfall under climate change: A case study in a tropical monsoon region with the climate projections from CMIP6 model. Journal of Hydrology 128002. doi: https://doi.org/10.1016/j.jhydrol.2022.128002

Author Response

Answer to Reviewer 3

Dear Reviewer,

We would like to thank you for reading, reviewing, and evaluating this work, which will undoubtedly contribute to improvements in the quality of our text. Thus, we present below the answers to the questions raised and some suggestions.

Comments and Suggestions for Authors

Major comments:

  1. What is the novelty of the work reported in this manuscript? The methods used in the study are all standard procedures used for similar kinds of studies in different regions of the world. 

Answer: Considering that extreme events are becoming increasingly frequent worldwide and that there is no work developed for the region studied using the climatic indices addressed in this study, this work will contribute to a better understanding of climatology, rainfall, and the occurrence of extreme hydrological events in the state of Pernambuco during the analyzed period. Thus, it helps in decision-making and strategies for regional adaptations in the face of climate change. 

  1. The assessment of extreme rainfall with the indices may not provide the actual representation. The authors need to add indices such as annual maximum rainfall of 1-day (and or 5-day) duration. This would provide further information about rainfall severity.

Answer: We are developing a new study where we will analyze other climatic indices in the same analyzed region to provide a more in-depth analysis of the study area and the severity of extreme events that occurred in the analyzed period.

Minor comments:

  1. Title should include the country name Brazil

Answer: Done.

  1. Page 1, last line: Authors should include more citations in the statement including recent studies such as (Das et al. 2022) and IPCC reports.

Answer: Done.

  1. What is the concrete outcome from the research- vulnerable to extreme rainfall/ drought?

Answer: As suggested by the reviewer, some adjustments were made throughout the text, where new information was inserted so that it was possible to answer the reviewer's questions and meet his suggestions. 

Reviewer 4 Report

It is my pleasure to review water-1769336 “A Recent Analysis of Rainfall Spatiotemporal Variability in Pernambuco State Using Climatic Indices” by Silva et al. The authors examined the spatiotemporal variability of rainfall in Pernambuco and the occurrence of extreme rain-fall events by climatic indices (including PCI, SPI, PCD, PCP and RAI) using the daily rainfall data from the historical series from 1990 to 2020. Extreme events have become increasingly frequent worldwide, this work will be helpful for a better understanding of the rainfall events that occurred in the state of Pernambuco during the analyzed period and provide data support for regional adaptive strategy under climate change.

Given the importance of this work and its potential impact, I would recommend that this paper is to be accepted by water once upon the following issues to be addressed.

Minor comments: There are many similar figures, it is recommended to integrate them into one, e. g. Figures 8-10; Figures 11-15.

Author Response

Answer to Reviewer 4

Dear Reviewer,

We would like to thank you for reading, reviewing, and evaluating this work, which will undoubtedly contribute to improvements in the quality of our text. Thus, we present below the answers to the questions raised and some suggestions.

Comments and Suggestions for Authors

Minor comments: There are many similar figures, it is recommended to integrate them into one, e. g. Figures 8-10; Figures 11-15.

Answer: As suggested by the reviewer, the number of figures in this article has been reduced to just one. They are grouped into a single figure.

Round 2

Reviewer 2 Report

I think the paper is appropriate to be published in Water. 

Author Response

Dear Reviewer,

Thank you for accepting our paper.

Best regards,

Authors

Reviewer 3 Report

The authors replied to my comments but did not mention the associated changes in the manuscript.  

Please mention the changes to the manuscript in the standard way e.g. page no. and line no. 

Otherwise, it is difficult to follow the changes in the manuscript. Highlighting some lines here and there is not enough.  

Author Response

Answer to Reviewer 3

Dear Reviewer,

We would like to thank you for reading, reviewing, and evaluating this work, which will undoubtedly contribute to improvements in the quality of our text. Thus, we present below the answers to the questions raised and some suggestions.

Comments and Suggestions for Authors

Major comments:

  1. What is the novelty of the work reported in this manuscript? The methods used in the study are all standard procedures used for similar kinds of studies in different regions of the world. 

Answer: Considering that extreme events are becoming increasingly frequent worldwide and that there is no work developed for the region studied using the climatic indices addressed in this study, this work will contribute to a better understanding of climatology, rainfall, and the occurrence of extreme hydrological events in the state of Pernambuco during the analyzed period. Thus, it helps in decision-making and strategies for regional adaptations in the face of climate change. (Page 3, Lines 91-103; Page 33, Lines 629-633)

  1. The assessment of extreme rainfall with the indices may not provide the actual representation. The authors need to add indices such as annual maximum rainfall of 1-day (and or 5-day) duration. This would provide further information about rainfall severity.

Answer: We are developing a new study where we will analyze other climatic indices in the same analyzed region to provide a more in-depth analysis of the study area and the severity of extreme events that occurred in the analyzed period.

Minor comments:

  1. Title should include the country name Brazil

Answer: Done.

  1. Page 1, last line: Authors should include more citations in the statement including recent studies such as (Das et al. 2022) and IPCC reports.

Answer: Done. (Page 1, Lines 39-43; 46-51).

  1. What is the concrete outcome from the research- vulnerable to extreme rainfall/ drought? 

Answer: As suggested by the reviewer, some adjustments were made throughout the text, where new information was inserted so that it was possible to answer the reviewer's questions and meet his suggestions. (Pages 33-34, Lines 639-660).
